# OpenReview forum: "The Neurosymbolic Frontier of Nonuniform Ellipticity: Formalizing Sharp Schauder Theory via Topos-Theoretic Reasoning Models"
_mathai.club/MathAI/2026/Conference — Submitted to 2026_

### Official Review · Reviewer_DUsN · 2026-03-12
**The Neurosymbolic Frontier of Nonuniform Ellipticity: Formalizing Sharp Schauder Theory via Topos-Theoretic Reasoning Models**

**Rating:** 5
**Confidence:** 3

**Review:**

Summary

The work presents a technical report describing the possibility of synthesizing two areas of modern science: the qualitative theory of nonlinear partial differential equations and the latest developments in the field of large reasoning models (LRMs) based on topos-theoretic frameworks and formal verification (Lean 4).
The author of the report begins with a detailed and mathematically competent exposition of the problem, describing the "Dark Side of the Calculus of Variations" and the key achievement of C. De Filippis and G. Mingione regarding gradient Hölder continuity. The text correctly explains the complexity of the problem and the technical innovation of the "ghost equation" method. The author then transitions to a description of LRM architectures, introducing the concepts of "thinking tokens," verification using Lean 4, and finally, a topos-theoretic interpretation of the logical inference process as the computation of a colimit in a slice topos. In the example, an AI model is applied to describe a method for deriving regularity conditions for a hypothetical log-multiphase functional.

Strengths

The very idea of linking pure mathematics (PDE theory) with AI and the foundations of mathematics (category theory, formal verification) is interesting for the automation of mathematical reasoning.
A fairly detailed analysis of the state of the problem and the contributions of various authors to the field is provided. The author clearly conveys the essence of the problem, the significance of the sharp threshold, and the complexity of the "ghost equation" technical solutions without delving into unnecessary details. References to key works and authoritative sources are provided. The logic of the narrative, moving from a specific mathematical problem to general AI methods, is traceable and supported by tabular material.

Weaknesses

Despite some strengths, the report lacks the full-fledged structure of a scientific publication: obtaining a specific new result, comparison with existing achievements of other authors and with other methods. Furthermore, the scientific novelty of the work—what exactly was accomplished by the authors—is unclear.

Recommendation

To present the material as a scientific work, it is recommended to revise the text and define the problem statement, describe the specific AI architecture that was used (or is proposed for use), and show how exactly the topos-theoretic constructions are algorithmically implemented and influence the inference process.

---

### Official Review · Reviewer_X9bb · 2026-03-13
**A part of a research proposal, not a paper.**

**Rating:** 3
**Confidence:** 4

**Review:**

The authors of the paper propose a refresher on the well-known idea of using formal proof assistants in combination with LLM to prove complex mathematical statements using Topos-Theoretic Reasoning. However, the text of the paper completely omits definitions of the new key concepts and methods for their use, making it impossible to assess the feasibility and complexity of the proposed approach. Although the article cites articles introducing these concepts, these articles are unpeer-reviewed preprints. The paper reads as an extended abstract for a project that has not yet been started, rather than a report on completed research. It would need to be fundamentally rewritten to include detailed, verifiable mechanisms and actual experimental results to be considered a credible scientific work.

---

### Official Review · Reviewer_fsF3 · 2026-03-13
**Incoherent fusion of various topics**

**Rating:** 3
**Confidence:** 4

**Review:**

This paper claims to synthesize recent results in nonuniform elliptic PDE regularity theory (De Filippis & Mingione 2025) with Large Reasoning Models (LRMs) and topos-theoretic AI architectures. It presents the sharp Schauder estimate $\frac{q}{p} < 1 + \frac{\alpha}{n}$ alongside categorical formulations of LRM reasoning (Eq. 7), proposing that AI systems can autonomously discover regularity bounds via "ghost equation methodology" verified in Lean 4.

**Strengths:**
- Clear and logical formatting
- Citation covers relevant work comprehensively

**Weaknesses:**
- Topics between Section 1 and Section 4 are not meaningfully integrated. LRM appears unrelated.
- Section 5 with working example does not provide any experiments to validate such system.
- Equation components are not defined (e.g. Eq. 3, 6, 7).
- No theorems, proofs nor experiments presented.
- Tables 1-4 are catalogues/timelines - no experimental data nor comparative analysis
- Appendix A list just claims without proofs; Appendix B is dimensionally problematic (colim of what and in which topos?)
- Popular science journalism, GitHub repos or blog posts do not qualify as a valid source for technical mathematics (Ref. 2, 16, 17, 19-20)

**Assessment:**
This paper does not contain results for mathematical or AI/ML hypotheses. It also lacks real experiments or meaningful integrations between concepts. In order to make this paper publishable, the reviewer recommends to:
1. Extend De Filippis-Mingione's results by providing original theorems with proofs.
2. Formalize the sharup Schauder estimate, and document the formalization process, libraries developed, and proof challenges.
3. Establish AI-assisted discovery track to assist in discovering regularity bounds.

---

### Decision · Program_Chairs · 2026-03-14

**Decision:**

Reject

**Comment:**

After careful evaluation by the Program Committee, we regret to inform you that your submission has not been accepted for presentation at MathAI 2026.

All submissions underwent a rigorous two-stage review process. Unfortunately, the reviewers identified one or more of the following concerns with your paper:

- Insufficient mathematical rigor or novelty relative to the existing body of work in the field;
- Presentation of results that substantially overlap with or rephrase previously published findings without clear original contribution;
- Significant issues with technical quality, including but not limited to broken or non-existent references, unsupported claims, or methodological gaps;
- Indications that the manuscript may have been generated with the assistance of large language models without substantial original intellectual contribution by the authors.

We received a large number of submissions this year, and the selection process was highly competitive. We encourage you to carefully consider the reviewers’ feedback (available through OpenReview), revise your work accordingly, and consider submitting an improved version to a future edition of MathAI or to another appropriate venue.

We appreciate your interest in MathAI and hope you will continue to engage with the conference community.

With kind regards,

MathAI 2026 Program Committee
URL: https://mathai.club
Telegram: https://t.me/MathAI_club
Email: mathai.club@yandex.ru